# How do Nucleotides Adsorb Onto Clays?

**DOI:** 10.3390/life8040059

**Published:** 2018-11-27

**Authors:** Ulysse Pedreira-Segade, Jihua Hao, Angelina Razafitianamaharavo, Manuel Pelletier, Virginie Marry, Sébastien Le Crom, Laurent J. Michot, Isabelle Daniel

**Affiliations:** 1Department of Earth and Environmental Sciences, Rensselaer Polytechnic Institute, Troy, NY 12180, USA; ulysse.pedreira@ens-lyon.org; 2Univ Lyon, Université Lyon 1, Ens de Lyon, CNRS, UMR 5276 LGL-TPE, F-69622 Villeurbanne, France; jihua.hao@univ-lyon1.fr; 3Laboratoire Interdisciplinaire des Environnements Continentaux, CNRS-Université de Lorraine, UMR 7360, 54501 Vandœuvre-lès-Nancy, France; angelina.razafitianamaharavo@univ-lorraine.fr (A.R.); manuel.pelletier@univ-lorraine.fr (M.P.); 4Laboratoire PHENIX, Sorbonne Université/CNRS UMR 8234, Case 51, 4 Place Jussieu, F-75005 Paris, France; virginie.marry@upmc.fr (V.M.); sebastien.le_crom@upmc.fr (S.L.C.); laurent.michot@upmc.fr (L.J.M.)

**Keywords:** adsorption, phyllosilicates, origins of life, geochemistry, surface chemistry

## Abstract

Adsorption of prebiotic building blocks is proposed to have played a role in the emergence of life on Earth. The experimental and theoretical study of this phenomenon should be guided by our knowledge of the geochemistry of the habitable early Earth environments, which could have spanned a large range of settings. Adsorption being an interfacial phenomenon, experiments can be built around the minerals that probably exhibited the largest specific surface areas and were the most abundant, i.e., phyllosilicates. Our current work aims at understanding how nucleotides, the building blocks of RNA and DNA, might have interacted with phyllosilicates under various physico-chemical conditions. We carried out and refined batch adsorption studies to explore parameters such as temperature, pH, salinity, etc. We built a comprehensive, generalized model of the adsorption mechanisms of nucleotides onto phyllosilicate particles, mainly governed by phosphate reactivity. More recently, we used surface chemistry and geochemistry techniques, such as vibrational spectroscopy, low pressure gas adsorption, X-ray microscopy, and theoretical simulations, in order to acquire direct data on the adsorption configurations and localization of nucleotides on mineral surfaces. Although some of these techniques proved to be challenging, questioning our ability to easily detect biosignatures, they confirmed and complemented our pre-established model.

## 1. Introduction

Clay minerals are ubiquitous on our planet as the result of low-temperature weathering or hydrothermal alteration of aluminosilicates and/or ferromagnesian silicates. They are also abundant in the matrix of many primitive chondrites, present in interplanetary dust particles, and have been described on Mars in the form of ferromagnesian smectite in early terranes from the wet Noachian Period, e.g., [1,2]. At the same time approximately on Earth, during the Hadean and early Archean, when life originated, no less than 33 different clay minerals have been described in a variety of settings with a large distribution in hydrothermal deposits [3]. As clay minerals have a strong affinity for organic molecules and can catalyze their reactions, e.g., [4], it has been proposed that they could have played an important role in the origins of life on Earth [5], whether life emerged at hydrothermal vents [6], at active serpentinizing sites, on dry land exposed to ultraviolet sunlight, or in evaporative environments [7]. In such different settings, the role of clay minerals is different.

Assessing the potential role of clay minerals in the origin of life requires some investigation of the environment of the nascent Earth, when it became habitable. This happened at the late Hadean, early Archean between ca. 4.1 [8] and 3.8 billion years ago [9,10], the age of the earliest preserved samples that might have retained an isotopic biosignature. This is particularly challenging as, by definition, there is no record of preserved sedimentary rocks from the Hadean. Therefore, we will not discuss the geological constraints on the chronology of the emergence of life and rather focus on the development of habitable conditions, which inform the experimental conditions for our work on the role of clay minerals in the concentration and polymerization of nucleotides.

Four billion years ago, the Earth had already witnessed the Moon-forming impact (4.5 Ga) and other energetic collisions. The early atmosphere of Earth was anoxic, likely dominated by CO_2_ and N_2_, and not by CH_4_ and NH_3_, as often assumed [11]. The partial pressure of CO_2_ had dropped down to a few tenths of a bar and was modestly above modern levels; correspondingly, the surface temperature had cooled down well below 100 °C, between modern values [12] and up to 50–60 °C [13]. The evaluation of surface temperature has been debated for the last two decades and still is as it depends on the method of evaluation. Most probably, the climate was clement when the first preserved samples showing evidence of life appeared in geological records [7,14]. The ocean was already well-developed and its volume in the early Archean was larger than that of today, possibly up to twice the size of the modern one [15] or most likely ¼ larger according to the most recent models [16]. While the atmosphere was essentially anoxic, methane ascended into the upper atmosphere, where photolysis liberated hydrogen that escaped into space. The extent of land that emerged in the early Archean may have been significantly less than at present, possibly limited to 10% or less [17]. As a consequence, more voluminous oceans would have flooded a greater proportion of the continental crust [18]. Most of the interactions between the ultramafic oceanic seafloor took place at a mean pressure of ~30 MPa, slightly lower than the present value. The ocean chemistry was very different from the modern one and this is critical when it comes to the interactions between the surface of clay minerals and the nucleotides, whose charge distribution strongly depends on pH. Just as for climate, there have been ongoing discussions on the pH of the early ocean over the last decades. The proposed pH estimates used to span from acidic to strongly alkaline. The most recent modelling study by Krissansen-Totton et al. [14] based on a geological carbon cycle model coupled with ocean chemistry calculated self-consistent histories of climate and ocean pH. The model predicts a temperate climate between 0 and 50 °C and a seawater pH for the early Archean between 6.3 and 7.2, in good agreement with the results of another model developed by Halevy and Bachan [19]. pH increased monotonically over Earth’s history to the modern value of 8.2. It is buffered by continental and seafloor weathering. Such a slightly acidic pH would have allowed the solubilization of ferrous iron and phosphate [20]. Estimates of the salinity of the ocean that witnessed the emergence of life have also been a subject of debate. Proposed values typically span between 1.2× to twice the modern seawater salinity, depending on the evaluation of the budget of modern evaporites [21,22] and reach some extreme values up to 15× the modern salinity [23]. The recent analysis of a large dataset of fluid inclusions in Archaean quartz samples from different localities indicates that the Archaean seawater appears to have had a chlorinity comparable to that of modern seawater [24] and definitely less than twice the modern value. However, there is little constraint on the detailed composition of the early Archean ocean. Considering reducing (due to moderate levels of H_2,g_) and acidic (high levels of CO_2,g_) surface environments on the early Earth and probably early Mars, the seawater was probably rich in Fe^2+^ and Mn^2+^, at least at millimolar levels leached from water-ultramafic rock interactions. At a pH around 6.5 in the early Archean—late Hadean as abovementioned, Fe^2+^ concentration, set by the solubility of amorphous greenalite, could have reached about 10 mM in the early seawater, like K^+^, Ca^2+^, and Mg^2+^ in modern seawater. Marty et al. [24] showed that the concentration of potassium in Archean seawater was very similar to the modern values of 10 mM (Cl/K = 50). Depending on when plate tectonics actually started, when granites, such as those in the oldest parts of Greenland, Canada, and Australia, massively emplaced and could be weathered, the level of the Na/Ca ratio of the early ocean could have been either low or high, like the modern Na/Ca ratio of 45. Prior to the emergence of continents, virtually no river delivered alkali elements to the oceans, while the hydrothermal alteration of the ultramafic or komatiitic seafloor delivered a significant amount of Ca^2+^ and Fe^2+^ to the ocean.

This review of the environmental conditions of the Earth during the late Hadean—early Archean, when life emerged, provides some constraints for our experiments that aim at understanding how clay minerals helped to concentrate and possibly polymerize some building blocks of life, namely nucleotides that were already available. Whether RNA or DNA came first is not central to the present study and we investigated the reactivity of both types of monomers on the surface of clay minerals. The abiotic synthesis of nucleotides, and the formation of nucleosides in particular, are highly debated and beyond the scope of the present contribution [25].

The review of the environmental conditions that prevailed in the early ocean indicates that we neither prefer nor vet some specific geological environments as the cradle of life. In all environments, there is a reference to clay minerals as a likely substrate to concentrate the otherwise dilute building blocks of life. Dry land appears as a relevant environment, as in such conditions, dehydration-hydration cycles that allow the polymerization of nucleotides occur [26]. Even in such environments, that represented less than 5% of the early Earth surface, clay minerals are invoked as they can intercalate organic molecules with positive or negative charges and protect them from decomposition by cosmic and/or UV rays [27,28,29]. In all other wet environments, it is relevant to investigate the mechanism of interactions between the clay surface and nucleotides in a common aqueous solution free of activating organics, as initially suggested by Bernal [5]. In the following, we present the effect of some selected ions on the adsorption of nucleotides on some clay minerals. The selection of minerals significant to the late Hadean—early Archean should be guided by the products of the hydrothermal alteration of an ultramafic or komatiitic rock, rich in iron and magnesium, and therefore producing (Fe, Mg) rich clay minerals, e.g., [30,31]. For instance, it includes serpentine minerals, nontronite, saponite, and chlorite; minerals whose occurrences have been reported in the geological record [3]. Surprisingly enough, most of previous studies have actually focused on Al-rich montmorillonite and kaolinite, with little characterization of the minerals’ surface, hence preventing further comparison between results, e.g., [32,33,34].

In the following, we present an integrated study on a series of well-characterized swelling and non-swelling clays in terms of their surface properties. Our multidisciplinary approach allows identifying the actual mechanism of nucleotides adsorption and the detailed effect of changes in the chemistry of the solution on both the mechanism and yield of adsorption.

This contribution reviews existing data on the adsorption of nucleotides onto phyllosilicates and additionally describes some complementary original data measured thanks to new techniques to characterize nucleotide-clay interactions. The first part of this contribution highlights the standardized method we developed and used to study the adsorption of nucleotides onto phyllosilicates, how our method differs from those presented in past studies, and how the newly proposed method allows insights into the mechanisms of interactions between nucleotides and phyllosilicates’ surface under various geochemical conditions to be gained. The second part presents original data exploring the use of in situ methods and reports some interesting outputs of theoretical simulations in aid of deciphering macroscopic experimental results. This present work is applicable to life’s emergence on the early Earth and to the search of molecular fossils or traces of life on other bodies of the Solar System.

## 2. Study of Adsorption in Aqueous Systems

### 2.1. Batch Adsorption Method

Measuring the adsorption of nucleotides onto clay minerals in aqueous solution corresponds to the study of the local enrichment in nucleotides at the solid/liquid interface compared to the bulk solution. Adsorption occurs and ultimately has to be described at the molecular scale, on an immersed surface in contact with the solution and accessible to the solute of interest. Although adsorption is caused by interactions and reactions happening at the nanometer scale, the consequences of adsorption can be observed in the bulk solution as it becomes depleted in the solute after equilibrium is reached [35].

Hence, one of the most common and convenient ways of studying adsorption at the solid/liquid interface is the batch (or immersion) method. In this method, solutions containing known concentrations of the solute are prepared over the studied concentration range. A fixed volume of each solution is poured and mixed with precisely weighted mineral samples. The samples are left to stand in the dark under controlled temperature conditions. After adsorption equilibrium is reached, the samples are centrifuged in order to completely separate the solid phase from the solution. The supernatant is retrieved and the equilibrium concentration of the solute of interest is measured. Using mass balance, the difference between the initial and equilibrium concentration of the solute allows for the calculation of the adsorbed quantity (in mole of adsorbate per g of mineral).

Adsorption isotherms depict an equilibrium relationship for the molecule of interest between the solid and the liquid phases, at a fixed temperature. Commonly, adsorption isotherms are displayed in graphs where the adsorbed quantity is plotted against the equilibrium concentration of the adsorbate.

Not only is the batch adsorption method easily carried out, it can also handily be adapted to various solid/liquid systems and physico-chemical conditions. For instance, one can study the effects of adsorbent properties such as solid size, available surface area, surface chemistry, or solid concentration; the effects of solution chemistry such as solvent nature (water, organic…), ionic strength, salinity, redox, or pH; the effects of solute properties; or the selectivity of the adsorbent surface towards mixed solutes.

As geochemical environments on the early Earth could have spanned a large range of physico-chemical conditions, as above-mentioned, our work has been focusing on understanding the effects of the chemistry of adsorbates and adsorbents, the pH, the salinity, and the temperature on the adsorption mechanisms of nucleotides on a variety of phyllosilicate minerals. We studied the adsorption of all eight ribonucleotides and deoxyribonucleotides onto a selection of geochemically relevant phyllosilicates [36,37,38]. Briefly, stock solutions of nucleotides have been prepared in the range of 0 to 3.5 mM. Most experiments were carried out using a seawater analog composed of 0.5 M NaCl and 0.05 M MgCl_2_. Moreover, the salt concentration of the solutions was also varied between 0.05 and 2 M and we tested numerous cations and anions: Na^+^, Li^+^, Mg^2+^, Ca^2+^, Zn^2+^, Ni^2+^, La^3+^ and Cl^−^, SO_4_^2−^, and HPO_4_^2−^. In most experiments, we let the samples equilibrate to a natural pH close to neutral; in specific experiments, we tested the effects of pH between 1.5 and 11 using the dropwise addition of concentrated HCl or NaOH. Mineral powders were prepared and checked for purity and homogeneity, and some of them were used as either coarse or fine grains [36,37,38,39].

For each experiment, a fixed volume of nucleotide solution was added to a known mass of mineral powder (1 mL total volume). Samples were then well-mixed and left to stand in the dark at room temperature for 24 h. Kinetic experiments already showed that adsorption equilibrium was readily reached in a few hours. Samples were then retrieved and centrifuged. The supernatant of controls and samples was analyzed using UV/vis spectrophotometry to measure the initial and equilibrium concentrations (Figure 1). Spectrophotometry was also used to detect the possible strong complexation of nucleotides to other solutes [38].

All adsorption experiments were conducted under ambient conditions of pressure and temperature, unless otherwise specified.

### 2.2. Limits of Macroscopic Studies

Adsorption at the liquid/solid interface is very different from gas/solid adsorption and cannot fully rely on the empirical models and methods developed for the latter. The simplest experimental setup to study adsorption in liquid solutions (i.e., mineral + solvent + solute) involves a binary solution interacting with the mineral surface. This implies that, compared to gas adsorption, (i) there is competition between the solvent and the solute for available sites on the mineral surface; (ii) hence, the composition of the adsorbed layer is unknown; (iii) statistically, the adsorbed molecules may exhibit a broader range of configurations [35]. The addition of salts in solution further complicates the study. Finally, in our case, the heterogeneity of mineral surfaces and the complexity of the nucleotide (molecular weight ca. 320 g/mol) certainly precluded any easy mechanistic approach using batch adsorption.

Phyllosilicates are layered minerals. Each unit is composed of various arrangements of tetrahedral and octahedral sheets (Figure 2). Particles have a high aspect ratio, i.e., a large basal surface area vs. small lateral surface area. While the basal surface is mainly hydrophobic, with siloxane groups and cavities, lateral surfaces are more oxide-like (Al, Mg, Fe), where pH-dependent hydroxyl groups interact with the solute. If the substitution of cations with a lower valence occurs in the crystal lattice of phyllosilicates, a permanent negative charge arises and is compensated by exchangeable cations in the interlayer space. This gives a swelling capacity to clays as the exchangeable cations retain shells of water and can be hydrated. Hence, while the study of adsorption onto non-swelling phyllosilicates deals with basal and lateral surfaces, the study of adsorption onto swelling clays must also account for the effect of exchangeable cations and the accessibility of the interlayer space. Moreover, phyllosilicate particles in solution have an organization at the micro- and mesoscales that is dependent on particle size, solvent chemistry, physical conditions, and/or solid concentration. They can form aggregates with intra- and inter-porosity and colloidal suspensions with small scale ordering. At the scale of the particle, they exhibit heterogeneous surfaces (basal, interlayer, and lateral) with very different reactive surface sites (chemistry, localization, and density).

Nucleotides are building blocks of life made of a phosphate group, a sugar moiety, and a nucleobase. The nucleobase is specific to each nucleotide: adenine (A), guanine (G), cytidine (C), thymine (T), and uracil (U). The sugar is different for ribonucleotides forming RNA and deoxyribonucleotides forming DNA, hereafter called NMP and dNMP nucleotides, respectively, where N stands for any nucleobase (Figure 3). Nucleotides exhibit a variety of configurations in solution or on the adsorbent surface and have several reactive groups (phosphate, amines, alcohol, and aromatic planes) that are sensitive to the physico-chemical conditions.

The overall complexity of the phyllosilicate-nucleotide-aqueous solution system requires consistent experimental and analytical methods and well-characterized minerals (with particular emphasis on surface properties) in order to compare studies. Otherwise, the search for a systematic understanding of adsorption mechanisms might be flawed.

## 3. Results of Batch Adsorption Studies

### 3.1. Half a Century of Data

Complexation of nucleotides onto clay surfaces has been investigated for decades [40]. The binding strength has been shown to be controlled by various factors: mineral surface, property of nucleotides, pH, salinity, and temperature, etc.

Clay minerals are composed of various surface sites, including -O(H), -Si, -Al, and metal sites. Metal sites primarily account for the adsorption of organics, including nucleotides [41]. The binding of nucleotides to homoionic montmorillonite is significantly enhanced when the exchangeable cations are transition metals [32,42,43,44,45,46]. Such stronger binding is ascribed to the higher complexation affinity of nucleotides, with transition metals compared with Na^+^ or alkaline earth metals (usually Ca^2+^ and Mg^2+^). At the microscale level, theoretical simulations have indicated that the binding is also dependent on the type of the surface (tetrahedral and octahedral) and the hydration of the surface [47]. For example, thymine and uracil are less stable on the tetrahedral surface of dickite (Al_2_Si_2_O_5_(OH)_4_) than on the octahedral surface. In this case, the hydrated octahedral mineral surface is where the most energetically favorable adsorption takes place.

Nucleobases, ribonucleotides, nucleotides, and their oligomers behave diversely on a given mineral surface, depending on the presence of functional groups and their relative complexation affinity. The surface complexation is mainly attributed to the phosphate group and ring structure of nucleobases. As a result, the adsorption of ribonucleosides and deoxyribonucleosides is lower than that of nucleotides, supporting the major role of the phosphate group in the adsorption of nucleotides [32,48]. Further evidence comes from the stronger preference of ATP over AMP on clay minerals [41]. In addition, the adsorption of nucleotides is totally suppressed on phosphate-treated clays [48]. The ring structure of nucleobases is usually positively charged at a low to intermediate pH and could complex with metals on the negatively charged clay surface. However, the perpendicular structure of nucleosides makes it difficult to interact with the minerals and consequently, nucleosides have lower adsorption than corresponding nucleobases [49]. When it comes to the oligomers of nucleotides, the length and secondary structure also affect their adsorption onto minerals. In similar experimental and theoretical studies on clays, it has been shown that the binding of oligonucleotides is dependent on the length and secondary structure [50,51,52]. The longer the oligomer, the stronger it binds to the mineral surface and double-stranded nucleic acids generally bind more strongly than single-stranded ones.

Nucleotides interact with mineral surfaces primarily through electrostatic forces. Since charges of the mineral surface and nucleotides vary with pH, the electrostatic force and binding strength vary accordingly. The clay surface is usually negatively charged over a large range of pH, but some nucleotides can be either positively or negatively charged, depending on the pH [37]. As a result, the adsorption of nucleotides onto Na, Mg, Ca-montmorillonite, and kaolinite displays a declining trend with elevating pH [40,48,50,53]. However, if the charge of some nucleotides/nucleosides such as uracil is insensitive to pH, their adsorption can remain almost constant across a wide pH range [50]. Furthermore, the addition of transition metals in the solution could modify the trend of pH effect, i.e., high pH promoted the adsorption of nucleotides [46,54]. This reverse trend is due to the formation of metal hydroxides on the surface of the clays, which modifies the surface properties of the clay [54].

Previous studies have explored the influence of total salinity on adsorption by showing either enhancing or hindering effects. For example, Ferris et al. [32] reported increasing in the binding of 5′-AMP onto montmorillonite from 2 to 11% by the addition of 0.2 M NaCl. In another example, Franchi et al. [55] investigated the adsorption of polynucleotides onto montmorillonite and displayed increasing adsorption in the presence of NaCl. However, Villafañe-Barajas et al. [49] revealed a detrimental effect of increasing salinity on the adsorption of 5′-AMP, adenine, and adenosine onto montmorillonite. X-ray diffraction patterns revealed that dissolved salts could occupy the interlayer space of montmorillonite and compete with organics for available sites [49]. Recently, Pedreira-Segade et al. [38] systematically examined the effect of salinity on the isotherms of nucleotides (dGMP) on swelling and non-swelling clays. At low concentrations of dGMP, increasing ionic strength promoted adsorption, whilst at high concentrations of dGMP, higher ionic strength either promoted adsorption on non-swelling clay or suppressed adsorption on swelling clay.

In addition to total ionic strength, the nature of salt significantly changes the adsorption of nucleotides onto the clay surface. Compared to NaCl, MgCl_2_ and CaCl_2_ enhanced the adsorption of nucleotides onto swelling clays, especially at high nucleotide concentrations [32,38,55], but whether this trend still holds true for non-swelling clays remains controversial. Franchi et al. [55] observed an enhanced adsorption of poly-nucleotides onto kaolinite in the presence of Ca and Mg compared with Na. Similarly, the presence of Mg and Ca could significantly enhance the binding of DNA or RNA onto the mica surface, whereas monovalent cations were less effective [56,57]. In contrast, Pedreira-Segade et al. [38] displayed a detrimental effect on dGMP adsorption in the presence of MgCl_2_ compared with NaCl. The controversy may lie in the different type or concentration of organics, i.e., polynucleotides and nucleotides, and requires further investigation. In a system with transition metal salts, the enhancing effect is even larger than for Ca^2+^ and Mg^2+^. For example, Hao et al. [54] showed that Zn^2+^ and Ni^2+^ significantly enhanced the adsorption of dGMP, dAMP, and AMP onto nontronite and montmorillonite surfaces compared to the effect of alkaline earth metals. As abovementioned, the presence of Zn^2+^ and Ni^2+^ in solution could also reverse the trend of adsorption at an elevated pH to greatly enhance nucleotide adsorption [46,54]. Furthermore, the enhanced binding of DNA and RNA onto mica in the presence of transition metals is widely acknowledged in astrophysical studies, e.g., [58].

Apart from metal cations, ions and some neutral compounds have been shown to alter the adsorption of nucleotides onto clays. For example, the addition of phosphate decreased the adsorption of ATP, ADP, and AMP onto montmorillonite, beidellite, and illite [41], and the adsorption of dGMP onto swelling and non-swelling clays [38], further confirming the pivotal role of the phosphate group in the complexation of nucleotides with a clay surface. H_2_S and thiosulfate were found to promote phosphate adsorption onto a pyrite surface [59]; whether this also applies for the adsorption of nucleotides onto clays is currently unknown. The presence of some organic ions could either promote the adsorption of nucleotides (acetate, [60]) or suppress the adsorption of nucleotides (citrate, [40,61]).

Similar to modern Earth, the prebiotic environment has a wide range of temperature and pressure conditions where early life might have originated. Life on modern Earth exists not only on the surface, but also in the deep subsurface, where temperature and pressure are higher than ambient conditions. There are very few investigations so far about the effect of temperature on the mineral adsorption of biomolecules; the effect is still debated or, honestly, unknown. For example, Goring and Bartholomew [40] demonstrated that increasing the temperature from 7 °C to 45 °C increased the rate of adsorption, but did not affect the amount of ribonucleic acid adsorbed onto bentonite, illite, and kaolinite. However, Graf and Lagaly [41] studied the adsorption of AMP onto montmorillonite, beidellite, and illite, and found an increased adsorption of AMP on the clays at 100 °C. Feuillie [62] investigated the adsorption of GMP and CMP onto various types of phyllosilicates at 4 to 95 °C and found an arched adsorption with increasing temperature, with optimal adsorption occurring at around 40–50 °C. More investigations would certainly be necessary to understand the temperature effect on nucleotides adsorption taking into account the stability of biomolecules at elevated temperatures and pressures.

In summary, numerous studies have examined effects of various environmental factors on the surface complexation of nucleotides on clays. It is currently hard to mutually compare these studies and reach any meaningful conclusions because they usually reported raw adsorption data that were not normalized to the specific surface area of adsorbents. Moreover, future experimental and theoretical works should really focus on conditions that are closely relevant to primitive Earth. For example, the early atmosphere and ocean were reducing due to moderate levels of H_2,g_, CH_4_, and CO. Under these conditions, the water-rock interaction would generate various reduced minerals, including Fe(II)-rich clays. These reduced minerals have never been investigated, probably because they are not stable under modern oxic conditions.

### 3.2. How Do Nucleotides Adsorb onto Clay Minerals?

As underlined in the previous section, past studies have tentatively proposed a large number of adsorption mechanisms, in positive correlation with the number of experimental conditions described for the adsorption of nucleotides and their polymers onto mineral surfaces. Unfortunately, only very few of these studies report quantitative data that allows for comparison between works.

The most common description and modelling of adsorption isotherms uses the Langmuir model, which was developed for gas/solid adsorption [63,64]. Although adsorption isotherms resemble Langmuir-type (I) gas isotherms, the Langmuir model cannot be applied for physical interpretations [65]. Indeed, nothing guarantees that a Langmuir-type isotherm in solution/solid systems represents the same type of mechanisms as described for gas/solid systems. As a reminder, the Langmuir model is based on the following hypotheses: homogeneous surface, gas adsorbate, formation of a monolayer, and no lateral interactions between adsorbates. Such ideal conditions are not guaranteed in the clay-nucleotide interactions and therefore the applicability of the Langmuir model is questioned. Indeed, a recent study pointed out that an increasing number of studies reported thermodynamic parameters using non-uniformized parameters in the Langmuir model, leading to large uncertainties and variabilities in the thermodynamic parameters of identical systems [66]! Instead, Azizian et al. [66] proposed a modified Langmuir model to approximate adsorption in a solid/liquid system, which temporarily allows for a universal calculation of the adsorption thermodynamic parameters by introducing a dimensionless constant in the equation.

Finally, in order to quantitatively compare the adsorption behaviors of nucleotides under various geochemical conditions, adsorption results have to be properly normalized. We summarize below our approach, normalization processing, and the main conclusions that have resulted from our previous work [36,37,38].

#### 3.2.1. Nucleotides Are Homologous Molecules

We have observed that adsorption isotherms of nucleotides on phyllosilicates under ambient conditions, in a seawater analog solution at pH 7, are asymptotic curves specific to each mineral-nucleotide pair analyzed (Figure 4a). However, for a given mineral surface, this does not mean that all nucleotides adsorb through different mechanisms or that a mineral surface exhibits a variable adsorption affinity for the nucleotides. Nucleotides only differ from each other through their nucleobase and the presence or absence of one hydroxyl group on the sugar moiety. As suggested in the vast literature on the adsorption of organic surfactants (often structurally close to each other), we proposed that the observed differential adsorption of nucleotides onto the same mineral surface could correspond to solubility differences. When the equilibrium concentration is normalized to the solubility of each nucleotide, adsorption isotherms of all nucleotides on a given mineral surface become similar (Figure 4b). This indicates that nucleotides are homologous molecules and adsorb via a common mechanism on a given mineral surface [36,37].

#### 3.2.2. Nucleotides Adsorb on Lateral Faces

Adsorbed quantities of a given nucleotide onto several mineral surfaces do not allow for direct comparisons: adsorbed quantities (in mol/g) are correlated to grain size of the powdered sample, i.e., a fine powder apparently adsorbs more nucleotides than a coarse one at a given equilibrium concentration (Figure 4c). Considering that adsorption is an interfacial phenomenon, the adsorbed quantity (in mol/g) should be normalized to the reactive surface area of the mineral (in m^2^/g) (Figure 4d). Only adsorption isotherms transformed into adsorption densities (in mole adsorbate/m^2^ surface area) can actually be compared (Figure 4d).

Phyllosilicate particles exhibit three very distinct surfaces, basal, lateral, and interlayer—in the case of swelling clays. For a given adsorbate, these surfaces might have different affinities or interactions with the nucleotide. Thus, the reactive surface area might not be equivalent to the total surface area of a particle. As previously described, a large number of studies have focused on the adsorption of nucleotides onto modified swelling clay and argued that adsorption occurred in the interlayer space of such clays. However, our group repeated the experiments and their characterization by X-ray diffraction, but showed that the interlayer distance of swelling clay particles did not expand at all upon the adsorption of nucleotides [36].

The comparison of adsorption densities between swelling and non-swelling phyllosilicates is thus an excellent way to further investigate the adsorption mechanism of nucleotides on clay minerals since the interlayer space of non-swelling phyllosilicates is not accessible to nucleotides [37]. We used the isotherms collected under ambient conditions in a seawater analog solution at pH 7, and normalized adsorption quantities to the total, basal, or lateral surface areas of minerals. Strikingly, the adsorption densities became similar for a given nucleotide on every mineral tested, after normalization to the lateral surface. In addition, the presence of a saturation plateau on the adsorption isotherms suggested the full coverage of the lateral surface by nucleotides [36,37].

Hence, our work showed that, under the conditions tested, the adsorption of nucleotides was localized on the lateral surfaces of swelling and non-swelling phyllosilicates indifferently, leading to similar adsorption density isotherms and saturation. Moreover, because nucleotides behave as homologous molecules on a given surface, we proposed that phosphate is the common reactive group of all nucleotides. Our work concluded that, at pH 7 under ambient conditions and in a seawater analog solution, nucleotides adsorb on the lateral surfaces of phyllosilicates via ligand exchange between their phosphate group and the metal hydroxyls of the lateral surfaces of the minerals (Figure 5) [36,37].

#### 3.2.3. The Effect of Extreme Conditions on Adsorption

Since early Earth environments encompass both acidic and alkaline environments, we tested the effects of pH on the adsorption of nucleotides in a seawater analog under ambient conditions. Our results showed that the adsorption of G-, C-, and A-bearing nucleotides onto swelling clays increased under very acidic conditions [36,37]. This is well-explained by electrostatic consideration as these nucleobases are positively charged below pH 4 and can interact with the negatively charged basal surfaces of swelling clay particles (no point of zero charge). The non-swelling phyllosilicates, however, have a point of zero charge between 2.5 and 4 pH unit, close to the pKa of those nucleotides and thus also become protonated below pH 4. This prevents any similar cooperative adsorption mechanism between G-, C-, and A-bearing nucleotides and non-swelling phyllosilicates. Under neutral to alkaline conditions analog to seawater and hydrothermal fluid, adsorption steadily decreased due to charge repulsion between the nucleotides and both basal and lateral surfaces. However, if the solution contained transition metals (e.g., Zn^2+^), the adsorption of nucleotides was strongly increased above pH 10 as large specific surface area metal hydroxyls precipitated and retained the organic molecules in solution [54].

The temperature of the early ocean that hosted the emergence of life was likely between 10 and 50–60 °C, though it is still hotly debated. Feuillie [62] tested the effect of temperature on the maximum adsorption of GMP and CMP onto swelling and non-swelling phyllosilicates in a seawater analog solution, at 1 bar, and found a slight optimum of adsorption at around 40–60 °C. Feuillie [62] also described an enhancing effect of temperature on the increased adsorption at low pH. Further work would be needed to interpret the latter result, as the combination of low pH and high temperatures might also favor the decomposition of nucleotides in solution.

#### 3.2.4. Ocean Chemistry Controls Adsorption

Despite the recent finding by Marty et al. [24] that the salinity of the Archean ocean was close to the modern one, the current understanding of the salinity of the Hadean ocean is still speculative. Consequently, we recently studied the effects of selected alkali and alkaline earth cations, transition metals, and molecular anions on the adsorption of dGMP (as a model) onto swelling and non-swelling phyllosilicates. We found that the reactivity of nucleotides is strongly modified by the salinity and composition of the solution [38].

Since salinity changes both the adsorption mechanism of nucleotides (chemical effect) and the structure and specific surface area of swelling clay particles (physical effect), we compared swelling and non-swelling adsorption isotherms and measured the structure of particles in situ with Small Angle X-ray Scattering (SAXS). This technique constrained the shape and size of swelling clay particles as salinity and nucleotide surface loading were changed. This powerful technique is often used to understand the organo-mineral interactions, e.g., the behavior of DNA-swelling clay colloids [67] or surfactants and phyllosilicates [68].

On one hand, alkali, alkaline earth, and transition metal cations complexed the phosphate group of nucleotides and this competition for the phosphate moiety inhibited the lateral adsorption via ligand exchange. However, alkaline earth cations also favored cationic bridging interactions on the basal surfaces of swelling clay minerals. Transition metals greatly enhanced adsorption compared to alkaline earth cations at pH ca. 7 due to higher affinity for the complexation of nucleotides [54].

On the other hand, molecular anions competed directly with the phosphate moiety of nucleotides for the adsorption sites on the lateral surfaces of clay minerals (metal-hydroxyls). Hence, their presence strongly decreased the adsorbed quantities of nucleotides measured for a given equilibrium concentration.

#### 3.2.5. Implications

To summarize, we have shown that nucleotides are homologous molecules that adsorb on the lateral faces (edges) of non-swelling phyllosilicates and swelling clays in NaCl-dominated solutions via ligand exchange. The addition of divalent cations (alkaline earth or transition metals) or molecular anions hinders this adsorption mechanism because the phosphate group is blocked through metal complexation in solution. Furthermore, metal complexation or low pH creates overall positive charges on nucleotides and thus allows for their adsorption onto the negatively charged basal surfaces of swelling clays through cationic bridges. Our current understanding suggests that swelling clays exhibit several adsorption mechanisms that might be optimal in certain geochemical environments, e.g., low pH high-temperature on-axis hydrothermal systems, high salinity evaporitic surface waters, or low temperature off-axis serpentinization systems. In Figure 5, we sketch our current understanding of the effects of environmental parameters on the adsorption of nucleotides onto phyllosilicates. The figure represents four different geochemical conditions and their effects on the mechanism of the adsorption of nucleotides (dGMP here is used as a model molecule) onto swelling and non-swelling phyllosilicates. The main adsorption mechanism of nucleotides, i.e., ligand exchange of the phosphate group on the metal hydroxyls of the broken edges of phyllosilicates, is modified and/or complemented with other mechanisms, depending on the conditions: (i) molecular anions at pH 7 adsorb on the mineral lateral surfaces and inhibit the ligand exchange process of nucleotides; (ii) below pH 4, in addition to ligand exchange, protonated nucleobases adsorb on the negatively charged basal surface of swelling clays; (iii) above pH 11, ligand exchange is drastically decreased on the edges of phyllosilicates but enhanced on the precipitated transition metal hydroxyls on the surface of clays; and (iv) the presence of divalent cations at pH 7 inhibits ligand exchange but favors the cationic bridging of nucleotides on the negatively charged basal surface of swelling clays.

Because the nucleotide-phyllosilicate system is very sensitive to the presence of salts, even in small amounts, we recommend that the study of their effect might be central to the understanding of the adsorption behavior of prebiotic organic molecules on minerals. Moreover, since nucleotides behave as homologous molecules when adsorbing onto phyllosilicates, we propose that one nucleotide could be used as a model molecule in future work. We are aware of the potential limitations that might arise, e.g., when it comes to generalizing low pH behavior or the possible aggregate formation of G tetramers at a very high concentration. Finally, the same conclusions can be drawn for the phyllosilicates tested. As adsorption does not depend on the chemical composition of phyllosilicates but primarily on swelling vs. non-swelling properties, we suggest that the study of one homogeneous sample of each structure might be sufficient to understand the system.

To go beyond the refined results of batch experiments and analyses, the next section explores promising new techniques adapted to the nucleotide-phyllosilicate system that might help acquire direct adsorption data.

## 4. Recent Developments and New Techniques

In adsorption systems as complex as those under investigation in the present article, it is particularly important to obtain confirmation of the proposed adsorption mechanisms using alternate experimental techniques or simulation procedures. Indeed, adsorption data, even on extremely well-characterized systems, only provide indirect information about the exact nature of adsorption mechanisms and adsorption sites. The following section details different attempts made to obtain such information.

### 4.1. Low Pressure Gas Adsorption

In terms of surface chemistry, natural solids are energetically heterogeneous and surface heterogeneity is intrinsic to real solid surfaces. Heterogeneity can arise from many different sources: different crystal faces, local crystalline disorder, surface reactivity, presence of impurities, and pores of various sizes and shapes. Clay mineral surfaces are heterogeneous, as they display at least two distinct surface environments with contrasting chemical properties, i.e., edge and basal surfaces; the role of which has been extensively discussed in the above sections. Numerous experimental techniques have been developed to analyze the surface heterogeneity of clay minerals and particularly to evaluate the relative amount of basal and edge faces or aspect ratio of a sample. Among those, low-pressure argon adsorption presents definite advantages compared to microscopic techniques such as AFM, as statistically relevant data can be obtained fairly easily. The principle behind aspect ratio determination by argon adsorption relies on the fact that adsorbate molecules fill surface sites according to the strength of the adsorbate-adsorbent bond. More energetic sites are filled at lower under-saturation (i.e., in a gas adsorption experiment, lower relative pressure) than less energetic ones. This dependence of adsorption on the nature of surface sites is particularly striking for monolayer adsorption that occurs at a very low relative pressure, typically P/P_0_ < 0.05, where P_0_ is the saturation pressure of argon. Analyzing surface heterogeneity by gas adsorption hence requires the analysis of adsorption at very low relative pressures, which can only be achieved using quasi-equilibrium set-ups equipped with accurate pressure sensors [69,70,71,72]. Experimental adsorption isotherms can be plotted as dV_ads_/d((Ln(P/P_0_)) vs. Ln(P/P_0_). The abscissa in such a representation corresponds to the free energy of adsorption in k*T* scale. It has then repeatedly been shown [70,71,72,73,74,75,76,77,78,79,80,81] that modeling the derivative as the sum of local derivative isotherms (Derivative Isotherm Summation (DIS) method) could yield a simplified image of surface heterogeneity. In particular, the DIS method provides the relative proportions of basal and lateral surfaces, when using a neutral gas such as argon.

As an example, Figure 6a displays the derivative adsorption isotherm obtained on a size-selected NAu2 nontronite. The DIS method allowed researchers to distinguish between lateral faces that adsorb argon at low relative pressures and basal faces that adsorb argon at higher P/P_0_. The exact assignment of the sites to structural features could be discussed since the amount of lateral faces appeared higher than the values obtained from TEM images. Figure 6b displays the derivative isotherm obtained after drying the initial NAu2 suspension conditioned in NaCl. The shape of the curve is similar to that observed for dry NAu2, but the adsorbed amounts are significantly lower. Such a change is probably due to a dilution effect, as upon drying significant, amounts of salts with a low specific surface area precipitated in the adsorption cell. This consequently decreased the total surface area of the samples. Still, it clearly appears that NaCl addition did not significantly alter the adsorption features, as far as adsorption energy is concerned. The situation is very different after dGMP has been adsorbed (Figure 6c). High-energy sites have vanished, which strongly points towards the adsorption of dGMP on the edges of the particles. We reached similar conclusions for pyrophyllite; edge faces did not appear any more as high-energy sites for argon adsorption after dGMP adsorption.

### 4.2. Vibrational Spectroscopy

Theoretically, vibrational spectroscopy could be an ideal tool to investigate the interactions and the bonding between minerals and nucleotides through their phosphate group [28,29,82]. Practically, infrared spectra show a strong overlap between the Si-O and P-O stretching vibrations at ca. 1000 cm^−1^. In adsorption experiments where the relative amount of nucleotide is orders of magnitude smaller than the amount of silicate, such an overlap precludes the detection of any small change in the wavenumber of the P-O stretching vibration due to M-O-P bridging, where M stands for all potentially hydroxylated metallic cations on the broken edges of the clay mineral, e.g., Si, Fe, Mg, and Al. Raman spectroscopy, which has a better spectral resolution, was therefore employed, using visible and UV excitation to show how spectra acquired by the Raman instrument of the instrument SuperCam of the Mars 2020 mission could inform on the nature of organics in the interaction with minerals, when looking for potential traces of early life. Figure 7 displays Raman spectra of dGMP, either dry or adsorbed at the surface of chlorite. At first, it shows that the characterization of the molecule requires that the laser power or energy is limited to ca. 0.5 Joules in the visible range to obtain the fingerprint of the molecule (Figure 7b). At higher energy in the visible range, the Raman signal retains only the signature of the P-O vibration and the guanosine molecule loses its structure and acquires the signature of a COHN aerogel, characterized by D and G Raman bands with an intensities ratio I_D_/I_G_ close to 1, e.g., [83]. With UV excitation, even at the lowest energy mandatory to measure the signature of dGMP, the Raman signal indicates that the molecule burns under the laser (Figure 7a) and acquires the characteristic signature of tholins with similar C/N and C/H ratios [84,85]. As shown in Figure 7c,d, the surface of chlorite may protect dGMP under specific conditions. The spectrum measured at 532 nm below ca. 1 J does not exhibit any band of burned organic matter or tholins. Unfortunately, the signal is also below the detection limit of dGMP in the sample. An increase of the incident energy to 3 J to go beyond the detection limit of such a small amount of dGMP leads to its degradation into amorphous COHN. When submitted to UV excitation, the Raman signal of dGMP adsorbed on the edges of chlorite is highly variable and cannot be explained at this stage. As a consequence, the search for organics on Mars using Raman spectroscopy could potentially lead to results limited to the presence or lack of organic matter, with little hope to decipher between different molecules.

### 4.3. High Resolution X-ray Absorption and Fluoresence

Further understanding of the detailed adsorption mechanisms deduced from careful batch adsorption data and summarized in Figure 5 now requires identification of the loci on clay surfaces where nucleotides adsorb, and therefore high spatial resolution techniques. The development over the last decade of synchrotron-based X-ray spectro-microscopy appears particularly promising to achieve this goal, as these techniques allow speciation at a sub-micrometric spatial resolution to be analyzed, i.e., in the range of particle sizes typical of the clay minerals used. Various techniques are currently available with hard (energy > 5–10 keV) or soft X-rays (energy < 3keV), with the latter being better suited for organics for three major reasons: (i) Soft X-rays allow access to the K-edges of numerous light elements from carbon (284 eV) to sulfur (2472 eV) and L-edges of numerous transition elements; (ii) at low energy below the K-edge of oxygen, in the so-called water window, water is almost transparent to X-rays and experiments on hydrated samples can then be performed; and (iii) focusing optics such as Fresnel lenses have become available in this energy range and allow spatial resolutions of ca. 20 nm to be reached. These three features appear highly promising to characterize the adsorption of nucleotides on clay mineral surfaces. 

For all these reasons, we used Scanning Transmission X-ray Microscopy (STXM) at beamline I08 at Diamond Light Source (Didcot, UK) to locate nucleotides adsorbed onto swelling and non-swelling phyllosilicate, at several equilibrium concentrations. In an STXM experiment, a Fresnel lens focuses incoming X-rays onto the sample, and both transmitted and fluorescence X-rays are measured. Controlled scanning of the sample then generates pixelated images at a given energy. By varying the energy, stacks of images are obtained at different energies, which yields spatially resolved chemical speciation data.

Figure 8 displays some of the fluorescence maps measured at an incident energy of 2300 eV for a 6 × 3 µm^2^ region and a pixel size of 200 × 200 nm^2^, on a dried pyrophyllite sample conditioned in seawater analog after the adsorption of dGMP at an initial concentration of 2.8 mmol/L. The maps of total signal (sum stack), silicon, and aluminium locate the clay particle in the field of view. The colocation of phosphorus with Si and Al shows that dGMP is adsorbed on a small pyrophyllite particle less than 1 µm in size. Despite the high spatial resolution achieved here, it is not yet high enough to locate the adsorbate on the particle edges. The presence of Mg and Na on these maps reveals the salts used in the experiments. These images were obtained for an incident energy of 2300 eV, just above the phosphorus K-edge, too far from the K-edges of oxygen and nitrogen (525 and 392 eV, respectively) to measure a meaningful signal for those elements, which have low fluorescence yields. Unfortunately, the signals of nitrogen and oxygen are close to noise and cannot be interpreted, although the distribution of nitrogen would have been interesting for the location of the nucleobase.

Figure 9a–c show STXM images and an example of the XANES spectrum obtained on a dried nontronite conditioned in a seawater analog solution after the adsorption of dGMP at an initial concentration of 1.2 mmol/L. The images (5 × 5 µm^2^) were obtained below and above the iron L-edge and the XANES spectrum was obtained across the Fe L-edge. Such XANES spectra clearly show the presence of the Fe-rich nontronite, but do not exhibit any clear spatial dependence of the signal. Therefore, identifying Fe adsorption sites from such an experiment is not feasible. Finally, Figure 9d–g present corresponding XANES spectra across the N K-edge for the same sample images (4 × 1 µm^2^) obtained at 410 eV and for two different regions of interest (ROI). The spectrum corresponding to the particle edges (Figure 9e) is significantly different from that measured on the bulk of the particle (Figure 9g). Furthermore, though noisy, the spectrum obtained from edge regions bears similar features to the reference spectra obtained on pure nucleotides [86].

These preliminary results clearly demonstrate the potential of synchrotron-based spectro-microscopic techniques for deciphering adsorption mechanisms in complex systems. In the case of clay minerals, exhibiting significantly different surfaces, they should allow adsorbed molecules to be localized and provide direct spectroscopic information on adsorbed species. Examining samples at various points of the adsorption isotherms should then reveal changes in the adsorption mechanism as the equilibrium concentration of adsorbate increases. Nevertheless, STXM experiments on adsorbed species require optimized protocols of sample preparation that were not achieved in our preliminary analysis. This is particularly relevant when working at very low energy around the carbon or nitrogen K-edge.

### 4.4. Theoretical Calculations

Computer simulations are nowadays extensively applied in various systems for refining adsorption mechanisms. Nevertheless, only a handful of simulations of nucleoside and nucleotide adsorption on clay minerals have been performed. Most of them considered the stability and/or polymerization of bases in the interlayer [27,87,88,89] and did not consider the fact that the interlayer of clay minerals is unlikely to be available for the adsorption in seawater, as shown in the previous sections. The few studies that examined adsorption on the external basal surfaces of clay minerals considered a low ionic strength environment [90,91]. 

In the context of the experimental evidence provided in the present contribution, it would be particularly worthwhile to simulate the adsorption of nucleosides and nucleotides on the edge faces of clay minerals. Despite the efforts to obtain reliable force fields for the edge faces of clay minerals [92], no truly satisfactory model is yet available for performing such simulations in a realistic way. Furthermore, as chemical reactions are involved, such a simulation would require the implementation of ab initio techniques. Since much effort has been devoted to experimentally analyzing the influence of cations on adsorption features on both the edge and basal faces, complementary molecular dynamics simulations could be performed in different salt solutions.

The simulations focused on GMP as the nucleotide. The clay model was a low charge cis-vacant montmorillonite with 0.75 charge per unit cell. Simulations were carried out in both NaCl and CaCl_2_ solutions at a constant solution charge, i.e., 0.5 M/L NaCl and 0.25 M/L CaCl_2_. The simulation box had a size of 41.44 × 35.88 × 62.5 Å^3^. The geometry of GMP was first optimized by Density Functional Theory (DFT). Its partial charges were calculated using the Restrained Electrostatic Potential (RESP) model [93] and the force field used was the General Amber Force Field (GAFF) [94]. The ClayFF force field [95] was used for montmorillonite, whereas water and ions were modeled using the SPC/E and Dang models, respectively. Molecular dynamics simulations were then carried out using the LAMMPS module [96] with an equilibration time of 50 ns. Trajectories were analyzed over 100 ns in the NVE microcanonical ensemble. The experiments were carried out twice for each system.

Figure 10 presents the time evolution of various GMP configurations in the NaCl-rich system. The black dots correspond to the state in which GMP is present in the system. In state 0, it is far from the surface. In state 1, it is adsorbed in a non-parallel configuration and in state 2, it is adsorbed in a parallel configuration. Based on a comprehensive inventory of trajectories, the molecule was considered to be adsorbed when distances between surface atoms and atoms from the molecules were ≤4 Å because interactions for a short amount of time due to translational trajectories are very low below this distance threshold. Snapshots in the right of the figure display some possible configurations characteristic of the various states. In the presence of NaCl, GMP spent most of the time in the solution during the course of the simulation. The relative duration of the adsorbed state versus the not adsorbed state was limited to ≈0.4.

The situation is markedly different with a CaCl_2_-rich solution (Figure 11). The duration ratio of adsorbed/free was significantly higher, between 2 to 5, even if the statistics did not allow an exact value to be provided. Such a high affinity of GMP for the montmorillonite surface is clearly due to the fact that the global charge of complexed GMP is higher in the presence of calcium than sodium (Figure 10 and Figure 11). This is well-illustrated in Table 1, which shows that whatever the considered state, the charge of GMP is twice to thrice higher in the presence of Ca^2+^ ions than that of Na^+^. Thus, Ca clearly enhanced the interaction of the nucleotide with the negatively charged montmorillonite surface.

These preliminary simulation results are in excellent agreement with observations of adsorption experiments that enhancing adsorption of nucleosides and nucleotides on the basal surface could occur through the formation of cationic bridges rather than mere charge screening. The observed differences between the Na- and the Ca-system could not be explained by a lesser amount of charges per molecule in the case of Na. In order to increase charge screening in a Na-system, a higher number of sodium ions should be placed close to the nucleotide. However, this cannot be achieved above a certain value due to steric hindrance.

## 5. Conclusions

Systematic adsorption studies using the batch adsorption method with well-characterized, homogeneous mineral samples allow for the treatment of adsorption isotherms, the derivation of adsorption densities, and the interpretation of adsorption mechanisms. The review of our and others’ work suggested that, in a seawater analog solution, at neutral pH and under ambient conditions, nucleotides behave as homologous molecules and mainly adsorb onto phyllosilicates via ligand exchange, as their phosphate group interacts with the metal hydroxyls of the lateral surfaces of phyllosilicate particles. This main mechanism can be inhibited with the addition of divalent cations, molecular anions, or changes of pH conditions. Upon modification, swelling clays exhibit a variety of secondary adsorption mechanisms that non-swelling phyllosilicates lack. For instance, at low pH, protonated nucleotides adsorb on the basal surfaces of swelling clay particles; or alkaline earth cations or transition metals complex with the phosphate group of nucleotides and act as cationic bridges on the basal surfaces of swelling clay particles.

This contribution also aimed at investigating the potential of new analytical and theoretical techniques for a detailed understanding of adsorption mechanisms of nucleotides on phyllosilicates in situ. Rather than staying speculative, a special effort was made to provide original data to assess the potential of each method. When the data could be interpreted, the results confirmed the adsorption mechanisms inferred from the batch adsorption method studies. Obviously, the list of techniques that was explored is not exhaustive. For instance, a recent study used Atomic Force Microscopy (AFM) and showed that lateral surfaces of phyllosilicates are more reactive to phosphate than basal ones; which points to the same results as our studies [97]. At this stage, in situ experimental techniques like vibrational spectroscopy or X-ray microscopy proved challenging and would require developments in the preparation of the samples. This suggests that the use of such techniques (vibrational spectroscopy) might not allow for the specific detection of biosignatures (i.e., discriminating between different organic molecules) in the search for the emergence of life on Earth and of traces of life on other planets, and should be approached carefully in future space exploration missions.

Prior to using demanding and expensive in situ analytical tools, the batch adsorption method is definitely an excellent and powerful starting point. Therefore, we encourage scientists working on the clay-water interface to take advantage of the proposed standardized method as the processing of adsorption isotherms requires an excellent knowledge of the physical and chemical properties of the adsorbent. This would indeed allow experimental results of different labs to be compared; the availability of distributed and interoperable data would allow the building of adsorption models to be applied to variable geochemical conditions, including, but not limited to, those that hosted the emergence of life on Earth and potentially elsewhere.

## Figures and Tables

**Figure 1 life-08-00059-f001:**
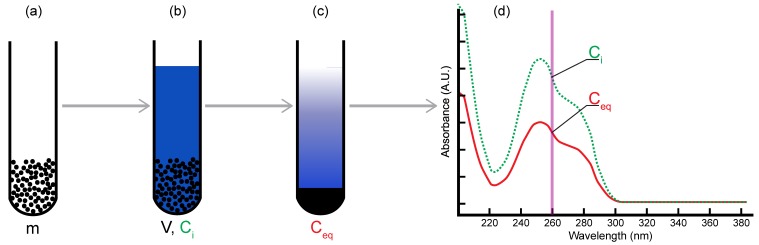
Batch adsorption method. Homogeneous suspensions or powders of minerals (**a**) are mixed with a nucleotide solution at a fixed temperature, pressure, pH, and solution chemistry (**b**). Samples are vortexed for 30 s and left to stand in the dark for 24 h. After equilibrium, they are centrifuged at 16,100 *g* for 25 min (**c**), and initial and equilibrium concentrations of nucleotides are measured with UV/vis spectrophotometry (**d**). See references [36,37,38] for more details on the preparation and characterization of homogeneous mineral samples. Adapted from [39].

**Figure 2 life-08-00059-f002:**
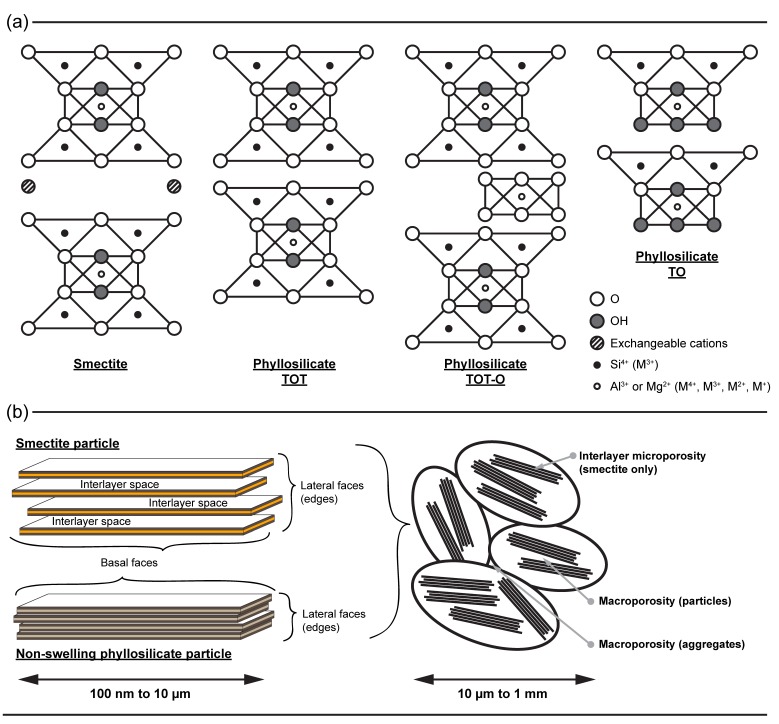
Structure of phyllosilicates: (**a**) Atomic arrangement of layers for various phyllosilicate structures; (**b**) ordering of phyllosilicate particles from smaller to larger scales. Adapted from [39].

**Figure 3 life-08-00059-f003:**
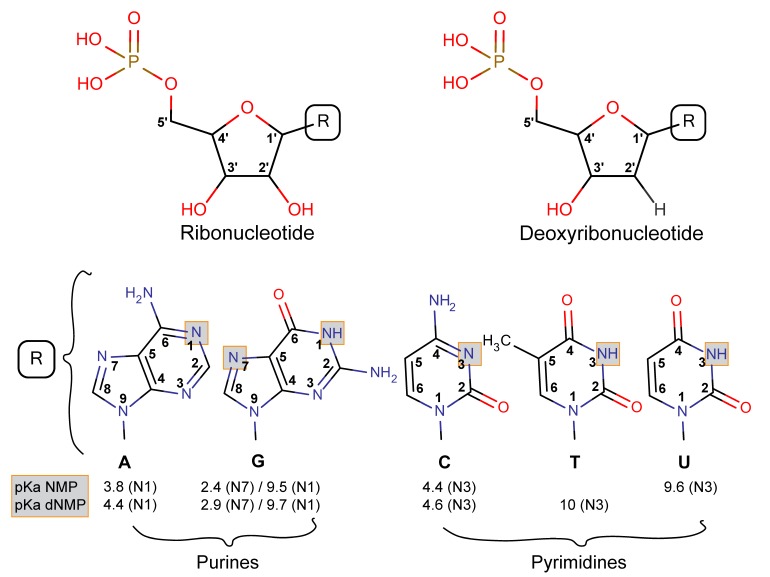
Structure ribonucleotides (NMP), deoxyribonucleotides (dNMP), and their nucleobases (R). The pKa of all nucleotides is also presented. Adapted from [37] with permission from Elsevier.

**Figure 4 life-08-00059-f004:**
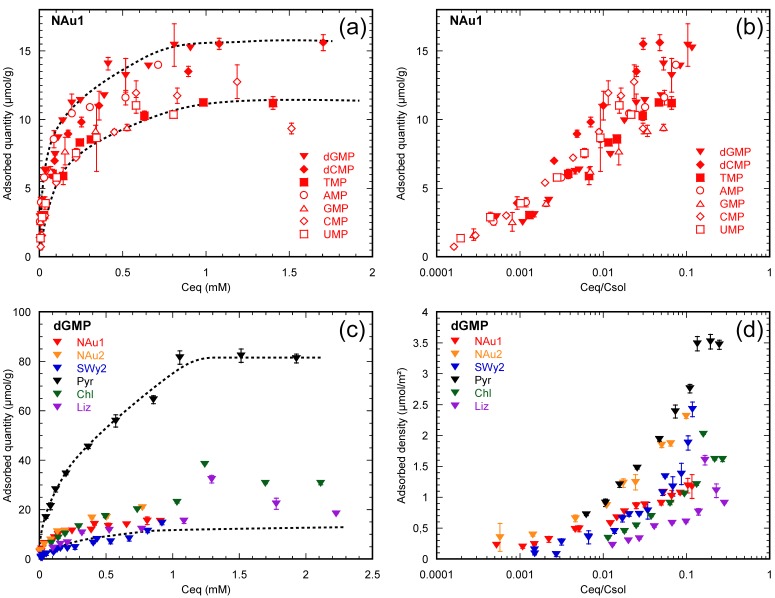
Illustration of the processing of adsorption isotherms: (**a**) Adsorption isotherms of all nucleotides on the same nontronite sample (Nau1); (**b**) same isotherms normalized to the solubility of nucleotides. Notice that all isotherms become similar and that differential adsorption vanishes; (**c**) adsorption isotherms of dGMP on three swelling clays (nontronites NAu1 & NAu2 and montmorillonite SWy2) and three non-swelling phyllosilicates (pyrophyllite (Pyr), chlorite (Chl), and lizardite (Liz)); (**d**) adsorption densities derived from the adsorption isotherms presented in the previous panel, normalized to the lateral specific surface area of the studied minerals. Adapted from [37] with permission from Elsevier.

**Figure 5 life-08-00059-f005:**
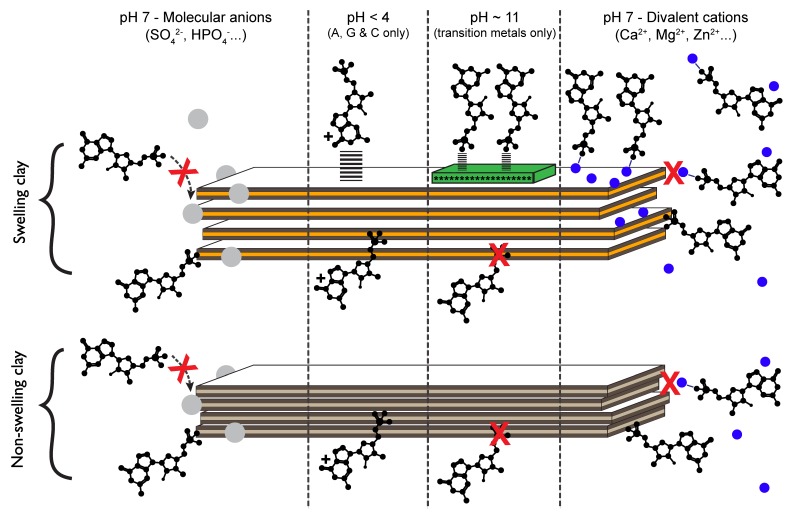
Adsorption mechanisms of nucleotides on swelling and non-swelling phyllosilicates illustrated under varying pH conditions and salinity. Blue circles represent divalent cations. Grey circles represent molecular anions. Green phase on the swelling clay surface at pH 11 represents the precipitation of metal-hydroxyls of low solubility (transition metals such as Zn^2+^). Adapted from [38] with permission from the PCCP Owner Societies.

**Figure 6 life-08-00059-f006:**
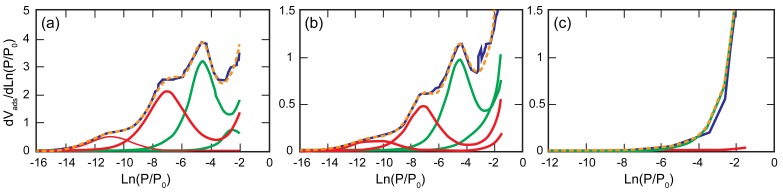
Derivative adsorption isotherms of the nontronite NAu2. (**a**) Raw sample; (**b**) sample dried in a seawater analog solution; (**c**) sample dried after the adsorption of dGMP in a seawater analog solution. Blue: Experimental data; Green: Basal domains; Red: Edge domains; Orange (doted): DIS fitting.

**Figure 7 life-08-00059-f007:**
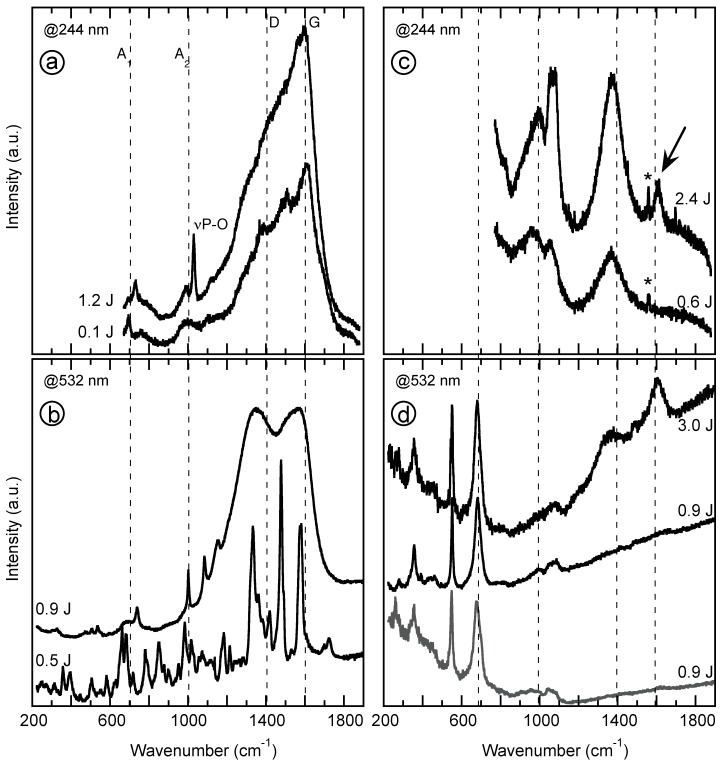
Visible and UV Raman spectra of dry deoxyguanosine monophosphate—dGMP (**a**,**b**), and dGMP adsorbed on the surface of chlorite (**c**,**d**). The spectrum displayed in grey (**d**) corresponds to the reference spectrum of chlorite without dGMP. * points to the stretching vibration of atmospheric O_2_ at 1536 cm^−1^. The dashed lines locate the characteristic bands of the breathing modes of aromatic/heterocyclic groups within the tholins (A1 and A2) and the bands observed for disordered carbonaceous materials (D and G).

**Figure 8 life-08-00059-f008:**
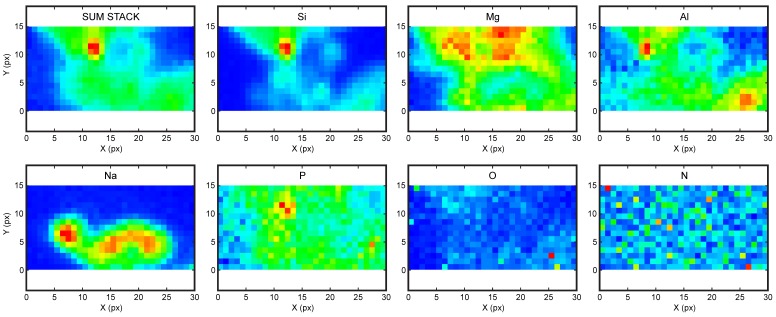
Fluorescence maps 6 × 3 µm^2^ measured at 2300 eV on a dried pyrophyllite sample saturated with dGMP.

**Figure 9 life-08-00059-f009:**
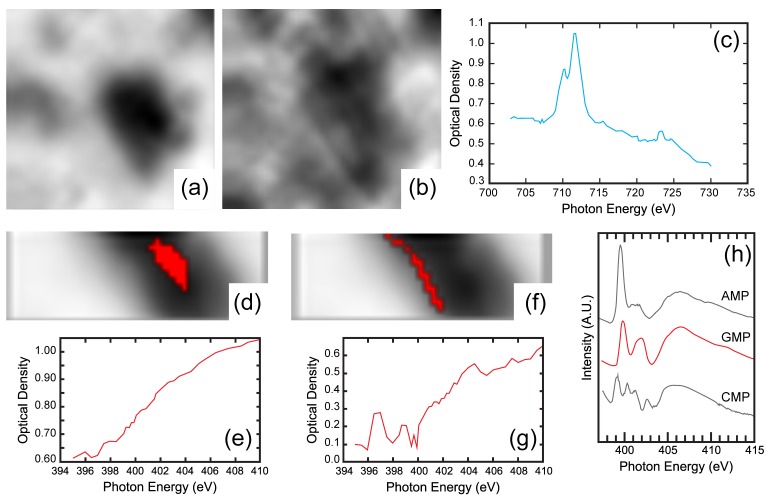
STXM images and XANES spectra of a dried nontronite sample saturated with dGMP. Images below (**a**) and above (**b**) the L-edge of Fe and corresponding XANES spectrum (**c**). Images and corresponding XANES spectra of an inner region of interest ROI 1 (**d**,**e**) and an outer ROI 2 (**f**,**g**) of the same nontronite particle saturated with dGMP. (**h**) XANES reference spectra for nucleotide films, data from [86]. The field of view is 5 × 5 µm^2^ in images (**a**,**b**), and 4 × 1 µm^2^ in images (**d**,**f**).

**Figure 10 life-08-00059-f010:**
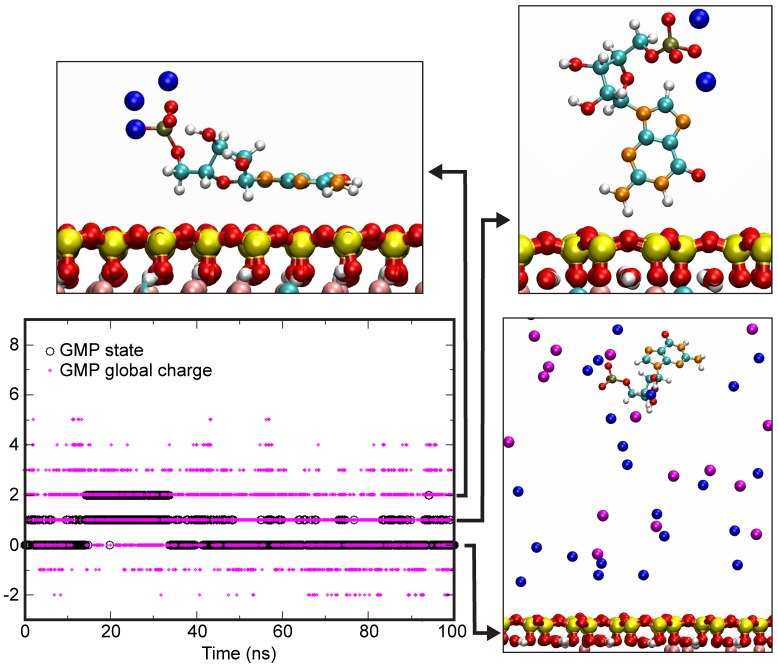
Left: Evolution of the global charge of GMP in a 0.5 mol/L NaCl solution and state of the GMP molecule with regard to the clay surface. Right: typical configurations of GMP molecules. Yellow: Si, pink: Al, red: O, white: H, cyan: C, orange: N, kaki: P, Blue: Na, Magenta: Cl.

**Figure 11 life-08-00059-f011:**
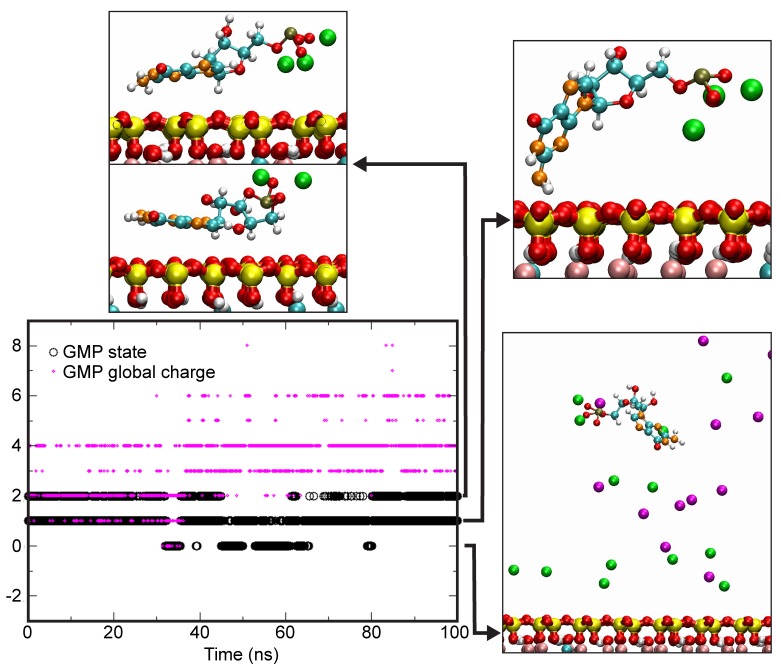
Left: Evolution of the global charge of GMP in a 0.25 mol/L CaCl_2_ solution and state of the GMP molecule with regard to the clay surface. Right: typical configurations of GMP molecules. Yellow: Si, pink: Al, red: O, white: H, cyan: C, orange: N, kaki: P, Green: Ca, Magenta: Cl.

**Table 1 life-08-00059-t001:** Charge of the GMP molecule in different states.

GMP State	Na^+^			Ca^2+^		
	Sim 1	Sim 2	Average	Sim 1	Sim 2	Average
0	0.92	0.79	0.85	3.44	2.56	2.92
1	1.42	1.41	1.41	3.60	3.70	3.64
2	1.09	1.56	1.31	3.74	3.17	3.38

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
