# Peer review of "How do Nucleotides Adsorb Onto Clays?"

_life, 2018, doi:10.3390/life8040059_

Reviewer 1 Report

This is a great paper, comprehensive, well-written and very important. English is fine. 

Author Response

Thank you for your review,

The manuscript has been modified based on the comments of reviewers 2 and 3.

Reviewer 2 Report

The contribution is scientifically very robust but needs some revisions before being published.

General comments:

I had difficulties to understand if this contribution was intended to be a review paper or an article describing new data. So, I recommend to state clearly in the introduction what kind of contribution is this.

The structure is not straightforward and hard to follow. I would recommend to describe the organization of the contribution at the beginning (in the introduction), so the reader does not get confused.

The contribution has implications both in the origins of life research and in the life detection investigations. I would suggest to structure the contribution better, by separating these two aspects and discuss the implications for these two investigations separately.

Specific comments:

Figure 1.
The legend you use here is not clear. What is 16, 100 g? Do you mean that you used for some experiments 16 g and for others 100 g? Please, clarify.

Then, did you evaluate the concentration based on the UV peak Intensity or area? From your figure it seems to me that you used the peak intensity at 260 nm. However, It would be much more accurate to measure the area of the peak between 220 and 300 nm. As described in Fornaro et. al. 2013 (International Journal of Astrobiology, 12 (1), 78-86), the estimation of the concentration from the peak area, instead of the peak height, is an improvement in the data analysis because, despite centrifugation, there can always be mineral particles in the supernatants causing scattering phenomena that could modify the shape of the peaks (distortions) and determine a shift of the maximum, and in these cases the evaluation of the concentration from the peak height could bring to erroneous concentration values.

For data analysis of UV spectra I would suggest to use a procedure based on a correlation between the concentration and the peak area. In particular, the conversion of peak areas (response) in concentrations requires the collection of data that establish the relationship between areas (or peak heights) and concentration of the solutions. These data are acquired with the calibration. The basic calibration procedure is called external standard method (ESTD):
                                          Concentrationx = RFx* Responsex
Responsex is the peak area for solution x, while RFx is the response factor, which is the factor necessary to convert the response measured from the UV spectrum in the concentration of the solution x. RFx is estimated establishing a function to express the concentration of solution x in terms of its response:
                                          RFx = f(Responsex)/Responsex
To determine the function f(Responsex), one or more solutions of known concentration of the substance to calibrate (standard solutions) are prepared. From the UV spectra you can measure the response caused by known concentration of component x in each standard solution and realize the calibration plot of Responsex vs. Concentrationx. The concentrations can be estimated from the peak height in the UV spectrum, i.e. the absorbance A, using the Lambert-Beer law (known the molar extinction coefficient ɛ tabulated at λmax). In fact, according to the Lambert-Beer law:
                                                          C = A/(ɛ * l)
If the calibration plot is a straight line you have:
                                                       RFx = 1/Slopex
                               Concentrationx = f(Responsex) = Responsex/Slopex
This is an interpolation method and from UV spectrum you can obtain the Responsex of the solution of unknown concentration and by interpolation derive the concentration.

Official methods of analysis provide invariably at least 3 calibration points to determine the linearity of the calibration plot, or 5 points in the case of non-linear calibration plots. Good Laboratory Practice (GLP) also require that the concentration of the compound used in the calibration should cover a range that includes the concentration of substance expected in the unknown solution.

Figure 5
Please, describe in detail in the text what you show in the figure.

Section 3.2, line 344-346
Please, explain better the modified Langmuir model.

Section 4.2, line 549-550
Please, rephrase for more clarity.

Line 563
Missing space between resolution and X-ray.

Line 603
Please, correct “bears”.

Line 631
Please, correct “as nucleotide”.

Line 692-694
I disagree. Based on my experience, you can detect organic signatures both using Raman and IR also at low coverage conditions for nucleotides adsorbed on clays. Did you measure just single spectra or did you acquire also maps?

Anyway, since your dataset is quite limited, I would suggest not to generalize

These revisions, I think, would improve considerably the clarity and impact of the manuscript.

Author Response

The contribution is scientifically very robust but needs some revisions before being published.

General comments:

I had difficulties to understand if this contribution was intended to be a review paper or an article describing new data. So, I recommend to state clearly in the introduction what kind of contribution is this.

This contribution presents both a summary of existing data on the batch adsorption method and new data on the application of new techniques to characterize the nucleotide-mineral interactions. The Special Issue on Minerals and the Origins of Life is the ideal opportunity to highlight our method, that is now well-developed and has proven robustness. This special issue is the definitely a good place to discuss thoroughly the method in the light of published (Feuillie et al. 2013; Pedreira-Segade et al. 2016, 2018; Hao 2018 under review) and unpublished data and to propose for the first time a comprehensive and synthetic interpretation of the effect of many environmental parameters that control adsorption of nucleotides on the surface of clay minerals.

A paragraph was added to the introduction to clarify the purpose of this contribution.

Moreover, the section 3 “Results” was renamed as “Results from batch adsorption studies”.

The structure is not straightforward and hard to follow. I would recommend to describe the organization of the contribution at the beginning (in the introduction), so the reader does not get confused.

A paragraph was added in the introduction to describe the organization of this contribution.

The contribution has implications both in the origins of life research and in the life detection investigations. I would suggest to structure the contribution better, by separating these two aspects and discuss the implications for these two investigations separately.

The conclusions were modified in order to present both views more clearly.

Specific comments:

Figure 1.

The legend you use here is not clear. What is 16, 100 g? Do you mean that you used for some experiments 16 g and for others 100 g? Please, clarify.

16,100 g is 16100 times the standard gravitational acceleration at the Earth's surface g.

Then, did you evaluate the concentration based on the UV peak Intensity or area? From your figure it seems to me that you used the peak intensity at 260 nm. However, It would be much more accurate to measure the area of the peak between 220 and 300 nm. As described in Fornaro et. al. 2013 (International Journal of Astrobiology, 12 (1), 78-86), the estimation of the concentration from the peak area, instead of the peak height, is an improvement in the data analysis because, despite centrifugation, there can always be mineral particles in the supernatants causing scattering phenomena that could modify the shape of the peaks (distortions) and determine a shift of the maximum, and in these cases the evaluation of the concentration from the peak height could bring to erroneous concentration values.

For data analysis of UV spectra I would suggest to use a procedure based on a correlation between the concentration and the peak area. In particular, the conversion of peak areas (response) in concentrations requires the collection of data that establish the relationship between areas (or peak heights) and concentration of the solutions. These data are acquired with the calibration. The basic calibration procedure is called external standard method (ESTD):

Concentrationx = RFx* Responsex

Responsex is the peak area for solution x, while RFx is the response factor, which is the factor necessary to convert the response measured from the UV spectrum in the concentration of the solution x. RFx is estimated establishing a function to express the concentration of solution x in terms of its response:

RFx = f(Responsex)/Responsex

To determine the function f(Responsex), one or more solutions of known concentration of the substance to calibrate (standard solutions) are prepared. From the UV spectra you can measure the response caused by known concentration of component x in each standard solution and realize the calibration plot of Responsex vs. Concentrationx. The concentrations can be estimated from the peak height in the UV spectrum, i.e. the absorbance A, using the Lambert-Beer law (known the molar extinction coefficient ɛ tabulated at λmax). In fact, according to the Lambert-Beer law:

C = A/(ɛ * l)

If the calibration plot is a straight line you have:

RFx = 1/Slopex

Concentrationx = f(Responsex) = Responsex/Slopex

This is an interpolation method and from UV spectrum you can obtain the Responsex of the solution of unknown concentration and by interpolation derive the concentration.

Official methods of analysis provide invariably at least 3 calibration points to determine the linearity of the calibration plot, or 5 points in the case of nonlinear calibration plots. Good Laboratory Practice (GLP) also require that the concentration of the compound used in the calibration should cover a range that includes the concentration of substance expected in the unknown solution.

We used the intensity at 260 nm in order to measure the concentration of nucleotides in solutions. As explained in Pedreira-Segade et al. (PCCP 2018), we used residual peak analysis with negative controls in order to check for residual mineral signal (interferences) in the analyzed samples. In addition, mineral interferences in supernatants are easily observed at low equilibrium concentration as they immediately induce an apparent negative adsorption.

We confirm that we used the ESTD calibration procedure, with carefully analyzed 10-15 points calibration curves measured throughout the experiments and repeated in the case of longer experiments to check for instrumental shifts.

Figure 5

Please, describe in detail in the text what you show in the figure.

The following text was added:

The figure represents four different geochemical conditions and their effects on the mechanism of adsorption of nucleotides (dGMP here is used as a model molecule) onto swelling and non-swelling phyllosilicates. The main adsorption mechanism of nucleotides, i.e. ligand exchange of the phosphate group on the metal hydroxyls of the broken edges of phyllosilicates, is modified and/or complemented with other mechanisms depending on the conditions: (i) molecular anions at pH 7 adsorb on the mineral lateral surfaces and inhibit the ligand exchange process of nucleotides; (ii) below pH 4, in addition to ligand exchange, protonated nucleobases adsorb on the negatively charged basal surface of swelling clays; (iii) above pH 11, ligand exchange is drastically decreased on the edges of phyllosilicates but enhanced on the precipitated transition metal hydroxyls on the surface of clays; (iv) the presence of divalent cations at pH 7 inhibit ligand exchange but favor cationic bridging of nucleotides on the negatively charged basal surface of swelling clays.

Section 3.2, line 344-346 Please, explain better the modified Langmuir model.

The sentence was modified because it could be interpreted as if we proposed a modified Langmuir model. Azizian et al. (Chemical Physcis, 2018) actually published the study presenting this novel adsorption isotherm description. The following text was added:

Instead, Azizian et al. [67] proposed a modified Langmuir model to approximate adsorption in solid/liquid system, which temporarily allows for universal calculation of the adsorption thermodynamic parameters by introducing a dimensionless constant in the equation.

Section 4.2, line 549-550 Please, rephrase for more clarity.

The text has been modified as requested.

Line 563 Missing space between resolution and X-ray.

The text has been modified as requested.

Line 603 Please, correct “bears”.

The text has been modified as requested.

Line 631 Please, correct “as nucleotide”.

The text has been modified as requested.

Line 692-694 I disagree. Based on my experience, you can detect organic signatures both using Raman and IR also at low coverage conditions for nucleotides adsorbed on clays. Did you measure just single spectra or did you acquire also maps?

Anyway, since your dataset is quite limited, I would suggest not to generalize.

We did not measure maps, but focused on repeating single spectra measurements. We used numerous different mineral samples and repeated analyses under carefully controlled conditions. We used adsorbed samples spanning the range of equilibrium concentration in nucleotides studied and focused our measurements on the ones having the maximum coverage reached, i.e. at saturation. This corresponds to a mass fraction of nucleotides ranging between 0.5-3 wt% in the mineral powders.

We used minimal sample preparation for vibrational spectroscopy analyses, thus trying to replicate natural outcrops potentially preserving organic signatures. The repeated one-point analysis instead of map measurements also better replicates current and future space exploration missions as rovers and probes generally do not measure maps. Although we did not try to optimize sample preparation for the in situ analyses, the small-size homogeneous mineral particles we used should theoretically help increase the adsorbed nucleotide density or abundance in the analyzed volume.

Although our data might be viewed as limited, we did not observe any clear signal of adsorbed, non-degraded nucleotides on the surface of phyllosilicates under the conditions we studied.

As far as we know, other studies such as Fornaro et al. (Icarus 2018) and Fornaro et al. (Astrobiology 2018) have used respectively higher mass fractions of nucleotides (ca. 5-6 wt%) and lower mass fractions (ca. 0.2-0.3 wt%). It would be interesting to try to replicate their analyses in order to understand what causes such discrepancies between all those results.

Based on our experience testing several vibrational techniques and synchrotron light source radiations on this kind of samples, we think that detection and analysis of adsorbed organic matter is not straight forward and should be approached carefully in space exploration missions.

We are aware that this might be an observation generating arguments so we modified the text.

These revisions, I think, would improve considerably the clarity and impact of the manuscript.

Reviewer 3 Report

Journal: Life (ISSN 2075-1729)

Manuscript ID: life-374177

Type: Article

Number of Pages: 27

Title: How do nucleotides adsorb onto clays?

Authors: Ulysse Pedreira-Segade , Jihua Hao , Angelina Razafitianamaharavo , Manuel Pelletier , Virginie Marry , Sébastien Le Crom , Laurent Michot , Isabelle Daniel *

Reviewer comments:

42           This sentence describes several distinct scenarios for which minerals could have “played an important role in the origins of life on Earth.” Alongside hydrothermal vents, serpentinization and photochemistry, I believe a reference of substrate interactions within thermally driven reactions in evaporative environments would be prudent to include. Corresponding examples of peptide systems could be found in (Orig Life Evol Biosph (2017) 47: 123. https://doi.org/10.1007/s11084-016-9516-z) or (ChemBioChem 2018, 19,1913 –1917. https://doi.org/10.1002/cbic.201800217)

304         Would add “the” before “presence” at the beginning of the sentence.

372         In section 3.2.2, regarding figure 4, the author describes the adsorption behavior across various minerals and nucleotides as “similar” when the isotherms are presented as normalized by either solubility or surface area (4.b and 4.d). I believe it would be prudent to quantify the degree of similarity.

411         It might be useful to include the PZC for the minerals studied here in some table or figure.

434         Where is the SAXS data presented and are the proceeding two paragraphs the results? This is unclear.

502         Does the “0.05” stated here need units? The sentence includes the term “relative pressure,” but the implication is unclear. If it is a unitless ratio of pressures, please clarify what are the items comprising the ratio.

538         I would change “investigated” to “employed” or a similar word.

591         I am unclear as to the major finding due to the STXM data in figure 8. While the intermediate signal green areas in Mg, Al and P are not discussed, it is unclear how a statement regarding adsorption of dGMP is inferred by colocation of Si, Al and P. Is it to be assumed that the data for O and N are homogenously noise, or is there importance to the isolated red pixels that show weak to no colocation with each other and do also not overlap with Si or P, but do possibly overlap with Al?

595         Figure 9 is presented in a difficult to parse manner; the labels a and b are not well applied to distinct sub-figures and the exact nature and subject of the sub-figures is unclear. I would suggest reworking and relabeling this figure with clarity in mind. 

602         How does the N-K XANES for ROI-1 compare to a negative control of the substrate or a blank?

609         Elaboration on future application of this technique for elucidating adsorption mechanisms would be ideal, joined with a reference, perhaps.

648         I think as a basic criterion for the condition being studied here, a reference for the vicinity to be considered adsorbed should be included. Could not that<0.4 nm vicinity be achieved for a brief amount of time, while a molecule is following some translational trajectory, and not result in a surface-bound state? Is there an additional time threshold for close vicinity that the molecules need to surpass to be considered adsorbed? If this is an unnecessary consideration, why? This should be addressed.

667         It is clear that the average charge for each state of the dGMP was higher in Ca simulations and a higher adsorption duration ratio was observed. While these experiments are preliminary, are you able to comment on the cause of the enhancement as due to a bridging effect best achieved with a divalent cation or due to the total amount of charge per molecule, which may be increased for Na simulations if a higher NA/dGMP molar ratio was investigated? I think this is worth addressing.

Overall, the author has presented a comprehensive and thoughtful consideration of nucleotide adsorption on clays. Much background and experimental context is included, which provides and motivation for the perspective taken, e.g. solubility and surface area normalization, and is very instructive. In general, more detail and discussion should be included for all experimental results. Results in section 4, especially, where a great diversity of techniques are applied to a model system, but the too brief discussions leave many questions unacknowledged or unaddressed.

I would recommend this manuscript for publication once the minor aforementioned revisions are addressed.

Author Response

This answer includes a Figure that seems to have disappeared from the page. In order to be sure that you receive it, we also uploaded a pdf document of the same answer.

42 This sentence describes several distinct scenarios for which minerals could have “played an important role in the origins of life on Earth.” Alongside hydrothermal vents, serpentinization and photochemistry, I believe a reference of substrate interactions within thermally driven reactions in evaporative environments would be prudent to include. Corresponding examples of peptide systems could be found in (Orig Life Evol Biosph (2017) 47: 123. https://doi.org/10.1007/s11084-016-9516-z) or (ChemBioChem 2018, 19,1913 –1917. https://doi.org/10.1002/cbic.201800217)

We agree with the reviewer that such an environment should be mentioned, even though clay minerals might not be as abundant in such a setting as in the others mentioned. Our sentence was intended to be as inclusive as possible of the various hypotheses for life’s emergence and hence it has been modified to:

As clay minerals have a strong affinity for organic molecules and can catalyze their reactions [e.g. 4], it has been proposed that they could have played an important role in the origins of life on Earth [5], whether life have emerged at hydrothermal vents [6], at active serpentinizing sites, on dry land exposed to ultraviolet sunlight or in evaporative environments [7 and references therein].

However, this first paragraph of the introduction is a general statement on clay minerals and nucleotides and we fail to see the necessity of adding references to silica or oxides minerals interacting with amino acids. We think that the review of Sleep 2018 and the references therein provide sufficient bases for this purpose of presenting the various hypotheses still debated in origins of life studies.

304 Would add “the” before “presence” at the beginning of the sentence.

The text has been modified as requested.

372 In section 3.2.2, regarding figure 4, the author describes the adsorption behavior across various minerals and nucleotides as “similar” when the isotherms are presented as normalized by either solubility or surface area (4.b and 4.d). I believe it would be prudent to quantify the degree of similarity.

The work described and summarized in this section has been validated in previous publications (e.g. Cases & Villiéras, Langmuir, 1992; Feuillie et al. GCA, 2013; Pedreira-Segade et al. GCA, 2016). Normalization to solubility and surface area were tested on all ribo- and deoxynucleotides and on swelling clays and non-swelling phyllosilicates, respectively. This method has also been successfully applied to alumina (Feuillie et al. Langmuir, 2015).

Below are log plots of the adsorption isotherms showed in Figure 4. We fitted a log regression curve to all the data and used the R² as a first approach to measure variance of the data before and after normalization. As can be seen, R² is increasing after normalization, suggesting a higher similarity of adsorption isotherms after the treatment has been applied. It can be noted that, in the case of the adsorption of dGMP onto all phyllosilicates, the R² stays quite low after normalization even if it increases compared to the raw data. This is probably a consequence of (i) natural sample retaining a fraction of impurities; (ii) intrinsic properties of minerals; (iii) analytical errors in the measurement of specific surface areas and uncertainties on the actual solubility of nucleotides. As discussed in our previous papers, even though the macroscopic approach of batch adsorption allows for the deciphering of first order interactions between nucleotides and minerals, these uncertainties are a limitation that has to be overcome through other techniques (e.g. the other methods described in this manuscript).

411 It might be useful to include the PZC for the minerals studied here in some table or figure.

A comprehensive figure as well as a complete list of the properties of the minerals prepared for these studies can be found in Pedreira-Segade et al. (GCA, 2016; Figure 7 and Table 2). Instead of adding a figure or a table for the PZC of minerals, we modified the text as follow:

“Our results showed that the adsorption of G, C and A bearing-nucleotides onto swelling clays increased under very acidic conditions [36,37]. This is well explained by electrostatic consideration as these nucleobases are positively charged below pH 4 and can interact with the negatively charged basal surfaces of swelling clay particles (no point of zero charge). The non-swelling phyllosilicates however have a point of zero charge between 2.5 and 4 pH unit, close to the pKa of those nucleotides and thus also become protonated below pH 4. This prevents any similar cooperative adsorption mechanism between G, C and A bearing nucleotides and non-swelling phyllosilicates.”

434 Where is the SAXS data presented and are the proceeding two paragraphs the results? This is unclear.

As stated by the other reviewer too, the presentation of our results is unclear. The introduction was modified in order to explain the fact that the results section presents a summary of the data collected and interpreted in previous studies using the classical batch adsorption technique. The SAXS data has been published recently in Pedreira-Segade et al. (PCCP, 2018) and is only briefly summarized here as this technique is complementary to the batch method in order to understand the behavior of clay suspensions in situ.

502 Does the “0.05” stated here need units? The sentence includes the term “relative pressure,” but the implication is unclear. If it is a unitless ratio of pressures, please clarify what are the items comprising the ratio.

The “0.05” is dimensionless as it is a relative pressure. It expresses the ratio of the pressure P in the sample cell over the saturation pressure P0 of the adsorbate gas at a given temperature.

For clarity, we modified the text as follow:

This dependence of adsorption on the nature of surface sites is particularly striking for monolayer adsorption that occurs at very low relative pressure, typically P/P0 < 0.05, where P0 is the saturation pressure of argon.

538 I would change “investigated” to “employed” or a similar word.

The text has been modified as requested.

591 I am unclear as to the major finding due to the STXM data in figure 8. While the intermediate signal green areas in Mg, Al and P are not discussed, it is unclear how a statement regarding adsorption of dGMP is inferred by colocation of Si, Al and P. Is it to be assumed that the data for O and N are homogenously noise, or is there importance to the isolated red pixels that show weak to no colocation with each other and do also not overlap with Si or P, but do possibly overlap with Al?

The XRF images were obtained for an incident energy located just above the phosphorus edge, i.e. at 2300 eV. Such an energy is far from the Kα edges of oxygen and nitrogen (525 and 392 eV, respectively). Furthermore, the fluorescence yields of low Z elements are rather low. As a consequence, for the incident energy used, the signals corresponding to nitrogen and oxygen are very close to noise and can then not be exploited. This was added in the text for clarity.

Moreover, the color coding of the maps is a relative abundance scale for each element and does not reflect absolute quantities. The comparison of STXM images and XRF maps allows us to clearly locate mineral particles and check their overlap with elemental abundance maps.

595 Figure 9 is presented in a difficult to parse manner; the labels a and b are not well applied to distinct sub-figures and the exact nature and subject of the sub-figures is unclear. I would suggest reworking and relabeling this figure with clarity in mind.

Figure 9 was redesigned for clarity and its caption was modified accordingly.

602 How does the N-K XANES for ROI-1 compare to a negative control of the substrate or a blank?

It is very close to a blank sample, which could strengthen the fact that adsorption occurs mainly on edges. However, as mentioned in the text, this result is, in our opinion, too preliminary to further elaborate on it.

609 Elaboration on future application of this technique for elucidating adsorption mechanisms would be ideal, joined with a reference, perhaps.

The text was modified for clarity.

These preliminary results clearly demonstrate the potential of synchrotron-based spectro-microscopic techniques for deciphering adsorption mechanisms in complex systems. In the case of clay minerals, exhibiting significantly different surfaces, they should allow localizing adsorbed molecules and provide direct spectroscopic information on adsorbed species. Examining samples at various points of the adsorption isotherms should then reveal changes in adsorption mechanism as equilibrium concentration of adsorbate increases. Nevertheless, STXM experiments on adsorbed species require optimized protocols of sample preparation that were not achieved in our preliminary analysis. This is particularly relevant when working at very low energy around the carbon or nitrogen K-edge.

Providing a reference is rather tricky as to our knowledge, the technique has not been extensively used for adsorption studies so far.

648 I think as a basic criterion for the condition being studied here, a reference for the vicinity to be considered adsorbed should be included. Could not that<0.4 nm vicinity be achieved for a brief amount of time, while a molecule is following some translational trajectory, and not result in a surface-bound state? Is there an additional time threshold for close vicinity that the molecules need to surpass to be considered adsorbed? If this is an unnecessary consideration, why? This should be addressed.

In fact, as mentioned by the reviewer, a criterion for distinguishing adsorbed vs non-adsorbed state based only on vicinity to the surface is not sufficient. Our analysis was based on a complete investigation of trajectories. It appears that situations in which the molecules are within 4Å of the surface for very short times are extremely rare. In most cases, when molecules are located close to the surface they remain there for some time before leaving the surface to go back in the solution. The initial sentence lacked details and has been changed in the revised version.

667 It is clear that the average charge for each state of the dGMP was higher in Ca simulations and a higher adsorption duration ratio was observed. While these experiments are preliminary, are you able to comment on the cause of the enhancement as due to a bridging effect best achieved with a divalent cation or due to the total amount of charge per molecule, which may be increased for Na simulations if a higher NA/dGMP molar ratio was investigated? I think this is worth addressing.

We do not think that the differences between Na and Ca could be totally erased by increasing the amount of sodium in the system. Indeed, charge screening (rather than bridging) plays a strong role in the system. To increase charge screening in the Na system would require placing much more sodium ions close to the nucleotide, which cannot be achieved sterically above a certain value. A more definite answer could be provided by carrying out simulations at higher NaCl contents. Still, we have added a sentence in the revised text to be more precise about how we "see" the system.

Overall, the author has presented a comprehensive and thoughtful consideration of nucleotide adsorption on clays. Much background and experimental context is included, which provides and motivation for the perspective taken, e.g. solubility and surface area normalization, and is very instructive. In general, more detail and discussion should be included for all experimental results. Results in section 4, especially, where a great diversity of techniques are applied to a model system, but the too brief discussions leave many questions unacknowledged or unaddressed.

I would recommend this manuscript for publication once the minor aforementioned revisions are addressed.

Round 2

Reviewer 3 Report

The understanding of the adsorption processes remain limited but the requested additions and revisions are generally addressed adequately so I see no reason to delay publication.